# The Role of AGE-RAGE Signalling as a Modulator of Gut Permeability in Diabetes

**DOI:** 10.3390/ijms23031766

**Published:** 2022-02-03

**Authors:** Matthew Snelson, Elisa Lucut, Melinda T. Coughlan

**Affiliations:** Department of Diabetes, Central Clinical School, Monash University Melbourne, Melbourne, VIC 3004, Australia; eluc0004@student.monash.edu (E.L.); melinda.coughlan@monash.edu (M.T.C.)

**Keywords:** advanced glycation endproducts, receptor for advanced glycation endproducts, diabetes, intestinal permeability

## Abstract

There is increasing evidence for the role of intestinal permeability as a contributing factor in the pathogenesis of diabetes; however, the molecular mechanisms are poorly understood. Advanced glycation endproducts, of both exogenous and endogenous origin, have been shown to play a role in diabetes pathophysiology, in part by their ligation to the receptor for advanced glycation endproducts (RAGE), leading to a proinflammatory signalling cascade. RAGE signalling has been demonstrated to play a role in the development of intestinal inflammation and permeability in Crohn’s disease and ulcerative colitis. In this review, we explore the role of AGE-RAGE signalling and intestinal permeability and explore whether activation of RAGE on the intestinal epithelium may be a downstream event contributing to the pathogenesis of diabetes complications.

## 1. Introduction

Diabetes mellitus has become one of the most problematic health concerns of the 21st century as the number of cases continues to multiply in both young and aged populations. As of 2015, it was estimated that the number of people affected by diabetes was 415 million, and is expected to rise to 642 million by 2040 [1]. Alarmingly, the number of people diagnosed with diabetes mellitus has quadrupled in the past 30 years [2]. There are two main forms of diabetes; type 1 diabetes (T1D) is characterised by an absolute insulin deficiency resulting from autoimmune destruction of pancreatic β-cells, whereas type 2 diabetes (T2D) is defined by a resistance to the actions of insulin, resulting in the reduction of glucose uptake and storage [3,4]. In both cases, hyperglycaemia is a defining feature and is the basis for the development of many diabetic complications [3]. Though an ageing population may partially explain the increased incidence of T2D, a number of environmental factors strongly increase the risk for development, particularly the overconsumption of diets high in refined sugars and fat, as well as lack of exercise [2,5].

Advanced glycation endproducts (AGEs) are a group of sugar-derived compounds commonly found in processed foods treated with dry heat. Their excess consumption is strongly associated with the pathophysiology of T2D [6]. Of concern, 57.9% of all calories consumed by the average individual in the US are derived from ultra-processed foods, from which 90% of all added sugar intake is derived [7]. Excess consumption of AGEs is also closely associated with many diabetic complications, including diabetic retinopathy [8,9] and neuropathy [10]. The receptor for AGEs (RAGE), has also been known for its implication in the development of long-term diabetic complications [11]. Recent evidence has shown that hyperglycaemia, a driver of endogenous AGE formation, can drive disruption of the intestinal barrier [12,13]. In this review, the mechanisms involved in promoting gut barrier dysfunction in diabetes, the formation of endogenous and exogenous AGEs, the effects of AGE-RAGE signalling in inflammation and disease, and the evidence regarding the disruption of the intestinal epithelial barrier by RAGE are examined and applied to the diabetic context and its associated complications.

## 2. Advanced Glycation Endproducts

### 2.1. The Maillard Reaction

AGEs are a group of compounds composed of modulated carbohydrate and protein subunits. These form when the carbonyl component of a sugar reacts with a free amino group of an amino acid in a series of nonenzymatic processes collectively known as the Maillard reaction (Figure 1) [14]. The Maillard reaction occurs in three distinct steps: initial, intermediate and final. The initial step involves a condensation reaction between the amino group of a lysine or arginine residue and the carbonyl group of a reducing sugar to form a Schiff base adduct during thermal processing. Subsequent rearrangement of the Schiff base results in a stable intermediate known as an Amadori product [15,16]. Amadori intermediates can be subjected to further reactions with a variety of other molecules, such as fragmentation, oxidation, enolization, dehydration and cyclization, to form more advanced products, including colourants, aromas and AGEs. For AGE precursors, degradation of intermediates from prolonged storage or high temperatures can result in α-dicarbonyls, which are highly reactive with their surrounding environment [16].

In the final stages of the Maillard reaction, these α-dicarbonyls can cross-react irreversibly with protein-bound arginine and lysine residues to form the resulting AGE. These reactions can be particularly damaging, as they are capable of altering the structure and potentially the function of endogenous proteins [16,17]. Some AGEs that have been well characterised include Nε-carboxymethyllysine (CML), carboxyethyllysine, pentosidine and pyrraline, which are used extensively to examine the physiological repercussions of AGEs, as is discussed later [17].

### 2.2. Endogenous AGEs

The presence of endogenous AGEs is increased in certain diseases, such as in age-related conditions or in diabetes, where excess circulating glucose favours AGE formation [16,18]. One perspective has suggested that increased AGE levels in serum can result from reactions of excess fructose and non-meat proteins in the intestines, not necessarily dietary AGEs, following observations that did not correlate with dietary AGE content [19]. Indeed, increased serum CML concentrations were detected in relatively healthy individuals consuming a high-protein diet [20]. Of most concern, AGEs and their precursors such as methylglyoxal, can alter the structure and function of endogenous proteins such that dysfunction arises at the cellular, tissue and organ levels [21,22]. The endogenous effects of hyperglycaemia were first identified in the blood of diabetic patients that showed excessive glycation of haemoglobin [23]. Following this, a positive relationship between the levels of blood glucose and AGE in the serum of diabetic patients was soon established [24], warranting an interest in further exploration of the pathological consequences of AGEs on human health.

### 2.3. Dietary AGEs

Cooking methods involving dry heat (grilling, baking, roasting, frying) can significantly increase the AGE content in food [25]. These practices are widely employed by food-manufacturing companies to enhance the flavour, appearance and aroma of food products [26]. Over the years, many databases have been generated that detail the AGE content in various food types, particularly foods consumed in the typical Western diet, using CML as a surrogate marker [27,28]. AGE content has previously been measured using a variety of techniques; however, ultra-performance liquid chromatography tandem mass-spectrometry is a preferred method due to its greater accuracy [16]. Even so, there is some discrepancy in the CML content of food groups determined by different studies, calling for more rigorous methods and study designs to be implemented.

In general, studies agree that the foods highest in CML content are dry goods such as breads, cereals and biscuits, as well as dairy products and meat dishes (Table 1) [27]. A later study confirmed that the foods with the highest AGE content included nut (peanut butter), or grain products treated at high temperatures, whereas butter, fruits, vegetables and coffee was relatively lower in CML content [28]. Substantial amounts of AGEs have also been detected in infant formulas, which raises concerns for the development of health issues at earlier stages of life [29]. A rise in the processed food sector in recent years has seen a greater number of individuals opting for more convenient, heat-treated food choices high in AGEs [26]. This correlates well with the increasing incidence of diabetes and other obesity-related complications [16,30]. Thus, when considering the changes in dietary intake over time, investigating the effects of AGEs in the development of disease has been of particular importance.

#### 2.3.1. Physiological Effects of Dietary AGEs in Diabetes: Animal and In Vitro Experiments

The consumption of dietary AGEs is thought to disrupt biochemical and physiological processes, including inducing inflammation, promoting cell death and altering insulin signalling [31]. In particular, the effect of exogenous AGEs in diabetes has been widely studied. Previously, it was observed that chronic AGE exposure, as glycated bovine serum albumin, in MIN6N8 cells and in pancreatic islets isolated from healthy mice, induced a T1D-like phenotype, including a dose-dependent reduction in glucose-stimulated insulin secretion, mitochondrial superoxide production, and increased glucose uptake [32]. A diabetic phenotype was also confirmed in vivo when rats were chronically exposed to high levels of AGEs, either through intraperitoneal injections or a diet high in AGEs. Of particular interest, it was revealed that an increase in plasma glucose only occurred in rats with high AGE exposure through diet rather than intraperitoneal injection, suggesting that the pathological effects of AGEs may occur mainly through processes occurring in the gut [32].

Similar effects were observed in Sprague-Dawley rats with chronic infusion of methylglyoxal, which presented with a T2D phenotype, including a decreased glucose tolerance in adipose tissue and glucose-stimulated insulin production by islets from the pancreas [33]. Upon further examination, it was found that exposure to high AGE levels blocked insulin production through the inhibition of its transcription factor Pdx-1 in rat pancreatic β-cell lines [33,34]. More importantly, these pathological effects could be reversed with a known AGE inhibitor such as alagebrium [32,33], or by blocking the major AGE receptor, RAGE, using anti-RAGE antibodies [34]. In a subsequent study, mice consuming a high glycaemic index diet for 12 months had increased insulin resistance and elevated serum AGE levels that correlated with increased progression of age-related macular degeneration. Interestingly, these effects were again diminished in mice that changed to a low glycaemic diet halfway through the study [35].

#### 2.3.2. Physiological Effects of Dietary AGEs in Diabetes: Human Studies

High AGE levels have been previously reported in renal failure patients, which correlated with AGE intake from the diet [36]. In this particular study, circulating AGE levels were measured before and after being randomized to a low or high-AGE diet. After 4 weeks, serum CML was significantly increased by 29% and serum methylglyoxal increased by 26% in patients on the high-AGE diet. A 2016 systematic review concluded that a high AGE diet was linked to increased levels of serum Tumour Necrosis Factor alpha (TNF-α) and AGEs in both healthy and chronically ill individuals, and that dietary AGEs may be involved in promoting certain chronic conditions, including T2D, cardiovascular disease and chronic kidney disease (CKD) [37].

As the literature in this field continues to expand, an increasing number of human studies continue to confirm the pro-inflammatory implications of dietary AGEs in a broad range of disease contexts. Early studies in diabetic patients have shown that high-AGE diets lead to an increase in AGE levels present in serum [38]. Furthermore, an increase in pro-inflammatory markers such as TNF-α, Vascular Cell Adhesion Molecule 1 (VCAM-1) and C-reactive protein (CRP) were also significantly increased [38]. Interestingly, changes to other inflammatory markers relating to endothelial dysfunction have also been observed in T2D patients. High AGE diets correlated with a significant increase in plasma E-selectin, Intercellular Adhesion molecule 1 (ICAM-1) and VCAM-1 only 2 h post-prandially [39]. However, the same effect could not be observed in healthy subjects after six weeks on a high-AGE diet [40]. Regardless, it was demonstrated that it only takes a single oral challenge with AGEs to induce acute endothelial impairment in both diabetic and non-diabetic participants [41].

Overall, the research into the dietary effects of AGEs on inflammation and vascular function has been limited due to lack of rigorous study designs and those of short duration [42]. One study looking at the association between AGE accumulation and lifestyle habits found a strong positive correlation between increasing age, waist circumference, consumption of meat products and cigarette smoking, and higher endogenous levels of AGEs [43]. However, the consensus is that further research needs to be conducted to investigate the detrimental effects of AGEs in both healthy and chronically ill populations to make more informed dietary recommendations [37,43]. The damage caused by AGEs, especially in diabetes, is often long-term and insidious, hence why it is important to understand the action of AGEs in different contexts for the development of targeted treatments and improved preventive strategies.

## 3. Receptor for Advanced Glycation Endproducts

Aside from crosslinking to endogenous proteins, it has been recognised that AGEs have a receptor that mediates cellular changes, inflammatory pathways and potentially, disease progression. RAGE is a transmembrane pattern recognition receptor belonging to the immunoglobulin gene superfamily. Many ligands are able to bind to RAGE, such as S100 proteins, nuclear material, and High Mobility Group B1 (HMGB1), and recognise three dimensional structures rather than a specific amino acid sequence [44,45]. RAGE is expressed on various cell types, including endothelial, neuronal cells and monocytes [44]. The receptor is commonly found in its full-length form comprised of a V-type domain, two C-type (constant) domains, a transmembrane and cytosolic region (Figure 2) [46]. RAGE also exists in two forms in the plasma: soluble RAGE (sRAGE), which is formed by cleavage of full-length RAGE by MMP9 and ADAM10, and an alternatively spliced version of RAGE referred to as endogenously secreted RAGE (esRAGE). These soluble plasma forms of RAGE lack the transmembrane and cytosolic regions but retain the capacity to bind to AGEs in serum. Rather than promoting inflammation, it is understood that sRAGE and esRAGE contribute to the clearance of AGEs, and thus prevent further insult to systemic tissues and end organs [47,48].

esRAGE is a decoy receptor produced by humans and has been recognised as a potential therapeutic in controlling AGE accumulation [47]. Indeed, esRAGE expression reduced the expression of proinflammatory and profibrotic genes when transfected into wild-type (WT) mesangial cells in vitro, whereas the opposite effect was apparent when full-length RAGE was overexpressed in these cells [49]. Importantly, these studies confirmed that RAGE in its full-length form promotes down-stream signalling of inflammatory events. Conversely, sRAGE and esRAGE are considered therapeutically beneficial due to their ability to reduce endogenous AGE accumulation.

### The AGE-RAGE Signalling Axis

As outlined in Figure 3, a variety of ligands, including free and protein-bound AGEs, S100 proteins, HMGB1 and genetic material, can bind to RAGE [50]. Following its activation, the stimulation of intracellular pathways such as the ERK1 and phosphatidylinositol-3 kinase pathway can lead to the activation and translocation of the nuclear factor kappa B (NF-κB) to the nucleus. Here, NF-κB transcribes a number of pro-inflammatory genes, including TNF-α, Interleukin 6, Monocyte Chemoattractant Protein 1, VCAM-1 and ICAM-1 [17,50]. As a result, inflammation and tissue injury can ensue in a variety of different settings. In diabetes, RAGE may also activate pathways that induce oxidative stress, including activation of NADPH oxidase to stimulate overproduction of mitochondrial superoxide [51,52] and a downregulation of mitochondrial complex I activity in the electron transport chain, which may inhibit the production of ATP. These events are especially important under hyperglycaemic conditions, where the over-stimulation of the RAGE receptor can trigger the release of excess superoxide from the mitochondria, a process that is associated with increased cellular damage and apoptosis [51]. Interestingly, activation of RAGE via AGEs increases the expression of RAGE on the surface of endothelial cells, as was observed in diabetic vascular complications [53]. Hence, RAGE activation in diabetes may have particularly damaging consequences in endothelial, epithelial and immune cells that can lead to pathological consequences, particularly in regions abounding in microvasculature, including diabetic nephropathy and retinopathy [50].

## 4. Function of the Intestinal Epithelial Barrier

The human intestinal barrier plays an important selective role in physically segregating the potentially harmful gut bacteria residing in the lumen from the host’s systemic tissues and organs, whilst still allowing dietary nutrients to be absorbed [54,55]. This barrier consists of a single layer of mucosal epithelial cells of various types that forms the innermost lining of the gut [56]. The integrity of the monolayer is maintained by a group of transmembrane proteins expressed between epithelial cells, named tight junction (TJ) proteins. Examples of TJ proteins include claudins (for example claudin-1 and 5), occludin, zonula occludens-1 (ZO-1) and tricellulin. Tricellulins are specialized structures which are able to act as tricellular tight junctions, where three cells contact one another [57]. In addition to the TJ proteins, adherens junction (AJ) proteins play a critical role in barrier integrity. E-cadherin is the main protein involved with AJ assembly, establishing contact between cells, which facilitates assembly of TJ proteins and subsequent barrier integrity [58]. To control the substances that may cross the gut barrier and into the systemic circulation, TJ proteins from adjacent epithelial cells complex together to strengthen the TJs and adhere to one another (Figure 4).

Gut permeability can be stimulated by different substances, including cytokines and bacterial toxins [59]. Importantly, gut barrier dysfunction is considered a major problem in diabetes, Irritable Bowel Disease (IBD) and other chronic illnesses [60]. One study that investigated the effects of hyperglycaemia on gut permeability in streptozotocin-induced diabetic mice found that there was increased dysfunction of the gut barrier, as evidenced by a reduced expression of ZO-1 protein [12]. Additionally, an increase in bacterial endotoxin was detected at systemic sites. Hence, a disruption of the regulative role of the gut barrier can allow unwanted substances, such as bacterial lipopolysaccharide (LPS), to enter the systemic circulation, trigger inflammation and modulate the immune system, which is linked to the development of a number of complications including CKD, Diabetic Kidney Disease (DKD), cardiovascular disease and neurological disorders [55,61].

## 5. RAGE as a Modulator of Intestinal Barrier Integrity

As a receptor for AGEs, RAGE is expressed widely throughout the gastrointestinal region. However, its expression is highly upregulated in diabetes, as was observed in both T1D [62] and less so in T2D [63] rat models. Morphological changes to the gut are well-documented, such as a thickening of the gut lining and smooth muscle, correlating with an increase in AGE and RAGE accumulation [62,63]. Yet, a mechanistic relationship in how the AGE-RAGE interaction influences the morphology and function of the diabetic gut must still be established.

HMGB1, a nonhistone nuclear protein that binds to RAGE, has previously been shown to disrupt the monolayer integrity of Caco-2 human enterocytic mononuclear cells and increase NF-κB DNA binding in both a time and dose-dependent manner [64]. Furthermore, hyperpermeability was successfully abolished when anti-RAGE or anti-p65 antibodies were applied, suggesting a key role for RAGE and NF-κB in the mediation of gut hyperpermeability events [64]. In mouse studies conducted by the same group, a significant increase in ileal mucosal permeability and bacterial translocation to the mesenteric lymph nodes in WT mice was absent in iNOS-deficient mice, further reinforcing the notion that gut permeability is dependent on increased oxidative stress in enterocytic cells [64]. Thus, it is evident that there are key inflammatory and pro-oxidative components that contribute to the disruption of the gut barrier in disease.

The association between RAGE and diseases of the gut has also been shown in both patients and animal models with IBD. One RAGE ligand, neutrophil-derived S100A12 (calgranulin C), has been shown to be elevated in bowel biopsies of patients with Ulcerative colitis (UC) and Crohn’s Disease (CD), which was associated with increased neutrophil migration, transient openings of intercellular junctions and increased NF-κB activation in the gut epithelium [65]. An association between RAGE activation and sustained NF-κB activation was further demonstrated in WT mice with a colonic injection of CML-enriched material, but not observable in RAGE null (*rage* −/−) mice [66]. In addition, increased RAGE expression and sustained NF-κB was only evident in inflamed areas of gut specimens from CD patients [66]. In 2007, Zen and colleagues demonstrated that RAGE expression was significantly elevated in intestine tissue sections from patients with active IBD, compared to healthy samples, using immunofluorescence labelling [67]. Using in vitro cell binding assays, RAGE also exhibited a role in transepithelial neutrophil migration by interacting with neutrophil cell-surface marker CD11b/CD18 [67].

Subsequent studies confirmed that elevated RAGE expression in the gut was only present in inflamed areas of gut samples, where CD or UC patients also had greater interleukin 1 beta (IL-1β) expression [68]. Furthermore, impeding the action of RAGE through the employment of the RAGE inhibitor FPS-ZM1 significantly decreased the severity and number of lesions in WT mice [68]. Whilst the role of RAGE in the diseased gut was defined, previous studies could not confirm the protection from intestinal inflammation in *rage* −/− mice [69]. However, this may be attributable to the fact that studies employed their own protocol and used different dosage to induce gut inflammation.

Previous in vivo and in vitro studies have also observed the ability for AGEs to influence expression of TJ proteins in the gut. In healthy Sprague-Dawley rats, histological examinations of the colon showed a change in the structure of the colonocytes with chronic exposure to excess AGEs [70]. However, and more importantly, high-AGE diets significantly decreased the expression of TJ proteins in the colon, such as occludin and ZO-1, compared to rats on a low-AGE diet. Contrary to previous findings [71], there was only a modest increase of LPS in the systemic circulation and no obvious signs of inflammation after prolonged feeding with a high-AGE diet. As the study duration only lasted 18 weeks, compared to 24 and 28 weeks for later studies, this suggests that the effects of AGEs and heat-treated (HT) diets in general have a gradual detrimental effect on the integrity of the gut barrier. In vitro, treatment of IEC-6 cells with glycated caseinate hydrolysates resulted in increased permeability and reduced tight junction expression in comparison to treatment with unglycated caseinate hydrolysates [72,73]. However, another study observed no significant difference in the monolayer permeability of Caco-2 cells when exposed to AGEs from infant formulas [29]. This may, however, be due to the fact that only two AGE concentrations were used, such that a range of AGE concentrations needs to be considered in order to confirm a true link of AGE exposure to epithelial integrity.

Recently, we demonstrated a link between intestinal permeability and increased kidney injury as a result of consuming a heat-treated diet high in AGEs [13]. In detail, Sprague-Dawley rats consuming a high-AGE diet had a significantly more permeable gut barrier as measured by a significantly altered expression of TJ proteins (claudin-1 and 5, occludin) in the jejunum, ileum and colon and an increase in serum LPS and complement activation. These effects were reversed with an AGE pathway inhibitor, alagebrium [13]. In a subsequent experiment, a db/db mouse model susceptible to developing CKD was fed either an unbaked standard chow or a HT diet high in AGEs. Unsurprisingly, diabetic mice on the HT diet showed a substantially more advanced progression of CKD, evidenced by worsening of albuminuria, and significantly greater disruption of the intestinal barrier compared to mice fed with a standard diet [13]. Hence, a causal relationship between the permeability of the gut and the progression of CKD and DKD is proposed, which may arise from the increase in inflammatory events following exposure to AGEs (Figure 5). However, more studies are required to examine the interaction between AGEs and gut processes, particularly in individuals susceptible to the progression of DKD and other complications.

It is evident that RAGE has a highly influential role in the modulation of inflammatory and immune responses, an attribute that could be exploited for the purposes of developing new targeted treatments, particularly for gut-mediated pathologies, where other therapies have been unsuccessful [68]. Given the apparent complex role of RAGE, however, this may create some controversy regarding its suitability as a therapeutic target. It is also worth mentioning that the literature regarding the mechanisms of action of RAGE in the diabetic gut is limited and under-researched, despite the fact that diseases such as diabetes, IBD and CD strongly correlate with increased gut permeability and the pathological effects of AGEs are well-established concepts [68]. These considerations are key when identifying potential RAGE-targeting therapeutics in different diseases.

## 6. AGEs and Gut Microbiota Composition in Disease Development

Typically, diabetic individuals also present with an altered gut microbial consortium which is associated with various metabolic dysfunctions including insulin resistance and increased gut permeability [74,75]. Several studies have also suggested the influential role of the gut and composition of intestinal microbiota in disease development when exposed to diets of varying AGE content in both mouse models and human subjects [35,76]. Microbiome and metabolomic studies in mice have identified a significant change in the composition of gut microbiota species and respective metabolite production in mice fed with either high or low-AGE diets, further reinforcing the implication of the gut as a conduit for disease [77]. A time-dependent effect on the variability of colonic microbes with long-term exposure to AGEs in healthy Sprague-Dawley rats was later confirmed [70]. Unlike rats fed on a low-AGE diet, beneficial short-chain fatty acid producing bacteria were reduced with a high-AGE diet after 18 weeks of feeding, whilst harmful bacteria were increased. Rats fed on a high-AGE diet also showed an increase in harmful gut bacteria and vascular dysfunction compared to rats fed on a low-AGE diet. Furthermore, these pathological effects were successfully reversed when the gut microbiome was suppressed using a broad-spectrum antibiotic [71]. Similar changes to the gut microbiota were observed in the study previously mentioned, where rats fed on a HT diet were shown to have expanded *Epsilonproteobacteria* and *Helicobacteraceae* taxa compared to rats on a standard diet, with increased LPS found in serum [13]. Thus, AGEs from an exogenous source may influence gut permeability through the microbiota in addition to direct effects from ligation with RAGE on the intestinal epithelium.

## 7. Conclusions

There is a growing awareness of the role of intestinal permeability in the pathogenesis of diabetes and its complication. Activation of RAGE plays an important role in the local inflammatory cascade in intestinal conditions such as Crohn’s disease and ulcerative colitis, and intestinal RAGE has been shown to be upregulated in the setting of diabetes. Recent evidence has demonstrated that exogenous dietary AGEs are associated with altered intestinal permeability, and this may occur via ligation of RAGE on the intestinal epithelium.

## Figures and Tables

**Figure 1 ijms-23-01766-f001:**
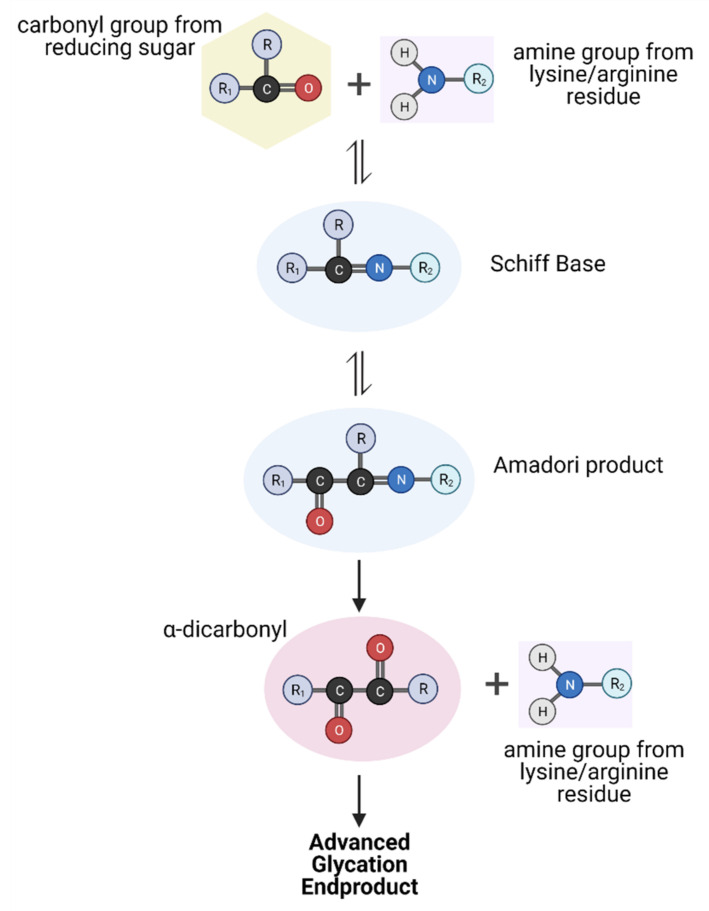
Overview of the Maillard reaction generating advanced glycation endproducts (AGEs). In the initial stages, the carbonyl group of a reducing sugar reacts with the amine group from an arginine or lysine residue to form a Schiff base. After a reshuffling process, the Amadori product is formed. Subsequent steps lead to formation of highly reactive α-dicarbonyls that can cross-link with endogenous proteins to form the resulting AGEs. Created with BioRender.com.

**Figure 2 ijms-23-01766-f002:**
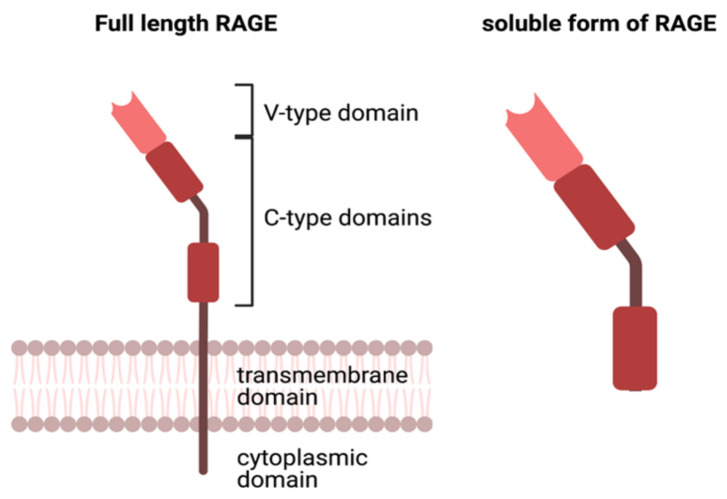
The structure of full length and soluble RAGE (Adapted from Lee and Park, 2013). Created with BioRender.com.

**Figure 3 ijms-23-01766-f003:**
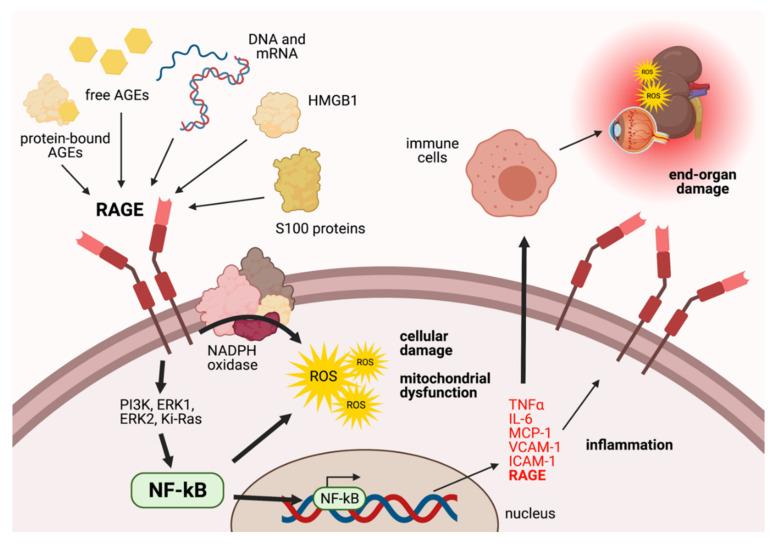
The RAGE signalling cascade (adapted from Gugliucci et al., 2014). RAGE is a multiligand receptor of the immunoglobulin gene family. Ligands include free and protein-bound AGEs, DNA, mRNA, High Mobility Group B1 (HMGB1) and S100 proteins. Upon ligation, RAGE can upregulate the overproduction of reactive oxygen species (ROS) via reduced nicotinamide adenine dinucleotide phosphate (NADPH) oxidase and activate a number of signalling cascades via phosphatidylinositol-3 kinase (PI3K), MAPK (ERK1 and 2), and Ki-Ras pathways to activate nuclear factor-κB (NF-κB). NF-κB can promote the production of ROS, ultimately leading to cellular damage and the dysfunction of mitochondria. NF-κB can also translocate to the nucleus to transcribe a number of pro-inflammatory cytokines and chemokines, including tumour necrosis factor α (TNF α), interleukin-6 (IL-6), monocyte chemoattractant protein-1 (MCP-1), vascular cell adhesion molecule-1 (VCAM-1) and intercellular adhesion molecule-1 (ICAM-1), which promote inflammation and stimulation of immune cells. RAGE expression is also increased. Hyperreactive immune cells can further exacerbate inflammation and ROS production, leading to further complications such as damage to microvasculature in the kidneys and eyes [50]. Created with BioRender.com.

**Figure 4 ijms-23-01766-f004:**
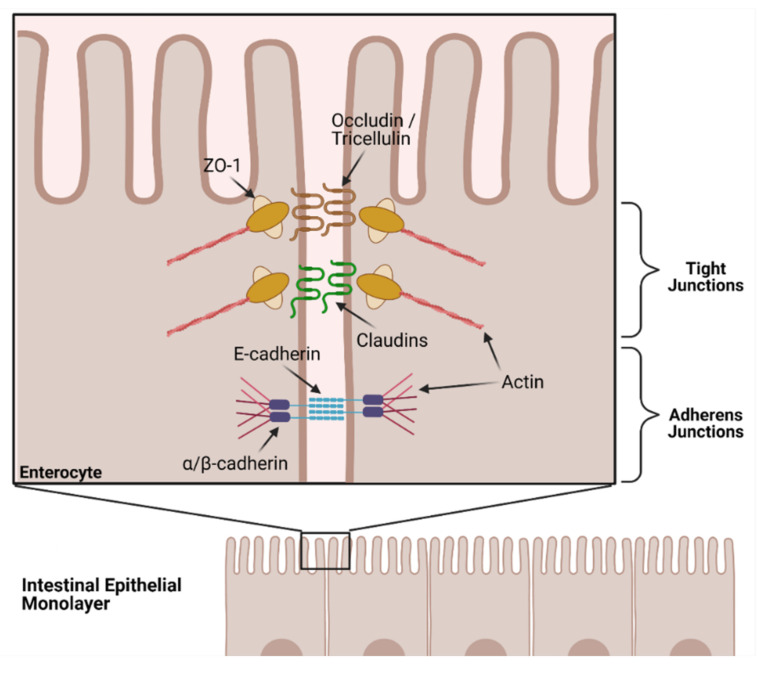
Schematic of tight junction and associated proteins from the intestinal epithelium (adapted from Suzuki, 2020). ZO-1 = Zonula Occludens-1.

**Figure 5 ijms-23-01766-f005:**
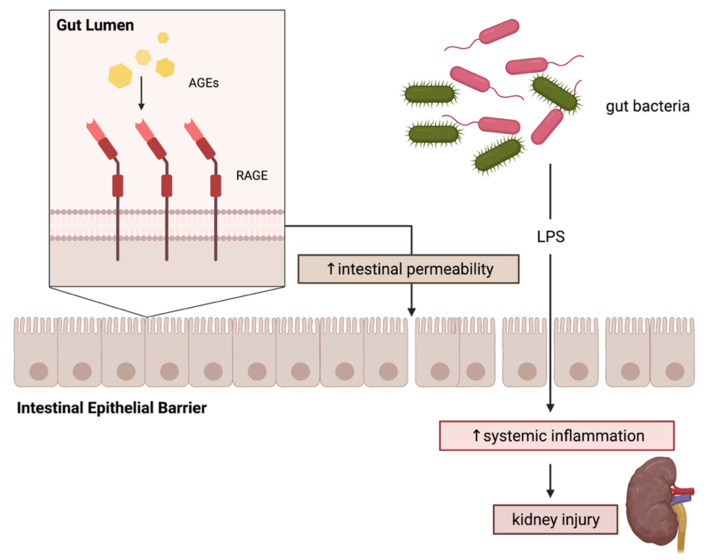
Proposed mechanism of RAGE-induced gut barrier dysfunction. Advanced Glycation Endproducts (AGEs) bind to the receptor for advanced glycation endproducts (RAGE) on the cell surface of gut epithelial cells to increase the permeability of the intestinal epithelial barrier. In turn, toxins from bacteria such as lipopolysaccharides (LPS) in the gut lumen are able to cross the barrier into the systemic circulation to trigger an inflammatory response, which may contribute to kidney injury. Created with BioRender.com.

**Table 1 ijms-23-01766-t001:** Average CML content in major food categories, expressed per mg/100 g food and mg/average portion size.

Food Category	mg CML/100 g Food	mg CML/Average Portion Size
Cereals	2.55	1.03
Meat dishes	2.42	4.81
Sweets and snacks	1.81	0.84
Bread and savoury biscuits	1.29	0.49
Meat and fish	0.86	0.9
Dairy products	0.44	0.36
Potatoes, rice and pasta	0.13	0.3
Fruits and vegetables	0.13	0.09

Adapted from Hull et al., 2012. CML = Nε-carboxymethyllysine.

## Data Availability

Not applicable.

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
