# Peer review of "The Role of AGE-RAGE Signalling as a Modulator of Gut Permeability in Diabetes"

_ijms, 2022, doi:10.3390/ijms23031766_

Round 1
Reviewer 1 Report
In this review article Snelson et al. summarize findings that argue for an AGE – RAGE signaling axis involved in the impairment of gut barrier function observed in diabetic patients. This review is nicely written and informative for a broad readership. However, the situation appears to be not as clear as presented in the manuscript. There is also evidence that glycation of e.g. caseinate at least partially protects from barrier damage. In addition, I have some other points that should be considered in a revised version of the manuscript.
Specific points:
- Line 54: Here I would expect to see the original reference of Maillard L.C. (1912) and not a reference citing an own review article.
- Figure 1: The legend should explain the meaning of R, R1 and R2. For readers new in the field of glycation and AGEs this is of importance. For R2 it is clear that it represents the amino acids lysine and arginine. R is depicted in the reducing sugar but is missing in the subsequent steps. As I understand the scheme, for the generation of dicarbonyl compounds (such as 1-deoxyglucosone or 3-deoxyglucosone) the lysine residue is released. In consequence it should be noted in the scheme as – amino acid instead of + amino acid. Moreover, if the amino acid is release, then R2 is lost and should not appear in the dicarbonyl species.
- Line 126: Here it would be interesting to see which type of AGE was used (AGE-BSA).
- Line 129: In reference 31, Fig. 2C it is reported that 2-deoxy-glucose uptake is increased not reduced.
- Line 194ff: The difference in generation of sRAGE and esRAGE should be explained.
- Line 249: The integrity of the monolayer is not only maintained by TJ-proteins but also by adherens junctions mainly in respect to mechanical stability. Impaired E-cadherin function also affects TJ structure and function. TJs are of central importance for barrier integrity.
- In respect to the discussed topic it might be interesting to also consider the articles of Wang SY et al. J Pharmcol Sci (2019) doi: 1016/j.jphs.2018.07.012; Sha H et al. Sci Rep (2018) doi: 10.1038/s41598-018-27787-2; Takino JI et al. Sci Rep (2021) doi: 10.1038/s41598-021-82619-0; Guibourdenche M et al. Toxics (2021) doi: 10.3390/toxics9060135.
- Figure 4 is a very simplistic view of tight junctions. In respect to barrier function specifically tricellular contacts should be mentioned.
Minor points:
- Line 40: References 8 and 9 should be combined [8,9].
- Line 46: delete “in”.
- Line 205: delete “is”.
- Line 214: Here, NF-kB
- Line 249: … innermost lining of the gut/intestine.
- Line 294: increase NF-kB
- Line 306: greater interleukin 1 beta
- Line 331: Introduce the abbreviation for “heat-treated” (HT)
- Legend Figure 5: Advanced glycation end products (AGEs)
- Line 367: Please rephrase the sentence “There have been studies have also suggested…
Author Response
In this review article Snelson et al. summarize findings that argue for an AGE – RAGE signaling axis involved in the impairment of gut barrier function observed in diabetic patients. This review is nicely written and informative for a broad readership. However, the situation appears to be not as clear as presented in the manuscript. There is also evidence that glycation of e.g. caseinate at least partially protects from barrier damage. In addition, I have some other points that should be considered in a revised version of the manuscript.
We appreciate the comments regarding the research in the area of caseinate glycation and barrier integrity. We note that caseinates (both glycated and unglycated) appear to protect against LPS-induced barrier dysfunction. However, when comparing glycated caseinate to unglycated caseinate, glycated caseinates have a reduced protective effect (that is to say glycation reduces caseinate’s protective effect against LPS-induced barrier dysfunction). It is an interesting point and we have added reference to these studies in the manuscript (Lines 336-338).
“In vitro, treatment of IEC-6 cells with glycated caseinate hydrolysates resulted in increased permeability and reduced tight junction expression in comparison to treatment with unglycated caseinate hydrolysates [72,73]”
Specific points:
- Line 54: Here I would expect to see the original reference of Maillard L.C. (1912) and not a reference citing an own review article.
We have updated to include the original reference.
- Figure 1: The legend should explain the meaning of R, R1 and R2. For readers new in the field of glycation and AGEs this is of importance. For R2 it is clear that it represents the amino acids lysine and arginine. R is depicted in the reducing sugar but is missing in the subsequent steps. As I understand the scheme, for the generation of dicarbonyl compounds (such as 1-deoxyglucosone or 3-deoxyglucosone) the lysine residue is released. In consequence it should be noted in the scheme as – amino acid instead of + amino acid. Moreover, if the amino acid is release, then R2 is lost and should not appear in the dicarbonyl species.
We apologise that the initial schema was unclear, and have updated accordingly. The reviewer is correct that during the formation of dicarbonyl compounds from Amadori products that the lysine residue gets realised. The previous version of this did try to show this (Amadori product ⟶ Dicarbonyl compound + Amino Group), though we recognise that this was illustrated confusingly. We have updated the illustration stating that it is R2 attached to the amino group, to make this release of the lysine/arginine clearer. Please see updated Figure 1 below:
- Line 126: Here it would be interesting to see which type of AGE was used (AGE-BSA).
Thank you for noting this. You are correct in thinking that this was AGE-BSA, we have included this detail in the manuscript.
- Line 129: In reference 31, Fig. 2C it is reported that 2-deoxy-glucose uptake is increased not reduced.
Thank you for noting this error, we have corrected this in the manuscript.
- Line 194: The difference in generation of sRAGE and esRAGE should be explained.
Thank you for this suggestion, the text has been updated to include reference to how sRAGE and esRAGE are generated (Lines 195-201).
“RAGE also exists in two forms in the plasma, soluble RAGE (sRAGE), which is formed by cleavage of full-length RAGE by MMP9 and ADAM10, and an alternatively spliced version of RAGE referred to as endogenously secreted RAGE (esRAGE). These soluble plasma forms of RAGE lack the transmembrane and cytosolic regions but retain the capacity to bind to AGEs in serum. Rather than promoting inflammation, it is understood that sRAGE and esRAGE contribute to the clearance of AGEs and thus prevent further insult to systemic tissues and end organs [46,47].”
- Line 249: The integrity of the monolayer is not only maintained by TJ-proteins but also by adherens junctions mainly in respect to mechanical stability. Impaired E-cadherin function also affects TJ structure and function. TJs are of central importance for barrier integrity. In respect to the discussed topic it might be interesting to also consider the articles of Wang SY et al. J Pharmcol Sci (2019) doi: 1016/j.jphs.2018.07.012; Sha H et al. Sci Rep (2018) doi: 10.1038/s41598-018-27787-2; Takino JI et al. Sci Rep (2021) doi: 10.1038/s41598-021-82619-0; Guibourdenche M et al. Toxics (2021) doi: 10.3390/toxics9060135.
We agree with the reviewer’s comments of the other components that are important in maintaining gut barrier integrity, we have included reference to these other components in our updated manuscript (Lines 252-257).
“Tricellulins are specialized structures which are able to act as tricellular tight junctions, where three cells contact one another [56]. In addition to the TJ proteins, adherens junction (AJ) proteins play a critical role in barrier integrity. E-cadherin is the main protein involved with AJ assembly, establishing contact between cells, which facilitates assembly of TJ proteins and subsequent barrier integrity [57].”
- Figure 4 is a very simplistic view of tight junctions. In respect to barrier function specifically tricellular contacts should be mentioned.
We have updated the figure to include reference to tricellulin, as well as the adherens junctions mentioned by the reviewer in the previous point. Please see the updated figure below:
Minor points:
- Line 40: References 8 and 9 should be combined [8,9].
- Line 46: delete “in”.
- Line 205: delete “is”.
- Line 214: Here, NF-kB
- Line 249: … innermost lining of the gut/intestine.
- Line 294: increase NF-kB
- Line 306: greater interleukin 1 beta
- Line 331: Introduce the abbreviation for “heat-treated” (HT)
- Legend Figure 5: Advanced glycation end products (AGEs)
- Line 367: Please rephrase the sentence “There have been studies have also suggested…
Thank you for noting these grammatical errors – they have been rectified in the manuscript.

Reviewer 2 Report
A review describes the role of advanced glycation products and their receptors in intestinal inflammation and permeability. Although much of the information presented in the manuscript is well known, the concept of the involvement of advanced glycation products in the development of diabetic complications through intestinal mechanisms is new. The manuscript is written in a clear and concise manner and can be published in its current form.
Author Response
We thank the reviewer for the comments and time taken to review our manuscript.
